# dbRUSP: An Interactive Database to Investigate Inborn Metabolic Differences for Improved Genetic Disease Screening

**DOI:** 10.3390/ijns8030048

**Published:** 2022-08-29

**Authors:** Gang Peng, Yunxuan Zhang, Hongyu Zhao, Curt Scharfe

**Affiliations:** 1Department of Biostatistics, Yale University School of Public Health, New Haven, CT 06520, USA; 2Department of Genetics, Yale University School of Medicine, New Haven, CT 06520, USA

**Keywords:** newborn screening, inborn metabolic disorders, tandem mass spectrometry, false positive screen, second-tier testing

## Abstract

The Recommended Uniform Screening Panel (RUSP) contains more than forty metabolic disorders recommended for inclusion in universal newborn screening (NBS). Tandem-mass-spectrometry-based screening of metabolic analytes in dried blood spot samples identifies most affected newborns, along with a number of false positive results. Due to their influence on blood metabolite levels, continuous and categorical covariates such as gestational age, birth weight, age at blood collection, sex, parent-reported ethnicity, and parenteral nutrition status have been shown to reduce the accuracy of screening. Here, we developed a database and web-based tools (dbRUSP) for the analysis of 41 NBS metabolites and six variables for a cohort of 500,539 screen-negative newborns reported by the California NBS program. The interactive database, built using the R shiny package, contains separate modules to study the influence of single variables and joint effects of multiple variables on metabolite levels. Users can input an individual’s variables to obtain metabolite level reference ranges and utilize dbRUSP to select new candidate markers for the detection of metabolic conditions. The open-source format facilitates the development of data mining algorithms that incorporate the influence of covariates on metabolism to increase accuracy in genetic disease screening.

## 1. Introduction

Newborn screening (NBS) has led to earlier life-saving treatment and intervention for more than 12,000 babies in the U.S. each year with a metabolic or another heritable condition, which has been recognized as one of the ten great public health achievements by the Centers for Disease Control and Prevention [1,2]. NBS starts with pricking a baby’s heel to collect a small blood sample, which is analyzed using tandem mass spectrometry (MS/MS) in order to measure metabolite levels, which are biomarkers for inborn metabolic disorders on the Recommended Uniform Screening Panel (RUSP) [3]. Blood metabolite levels are influenced by several continuous and categorical variables such as gestational age (GA), birth weight (BW), sex, ethnicity, age at blood collection (AaBC), nutritional therapy, and season of birth, which has been shown to affect the accuracy of screening [4,5,6,7,8,9,10,11,12,13,14]. To improve screening performance and separation of true and false positive cases, metabolite levels can be adjusted in relation to these covariates using post-analytical interpretive tools in Clinical Laboratory Integrated Reports (CLIR, formerly R4S) [15,16,17]. Recently, machine-learning-based approaches have been developed to identify hidden patterns in metabolic screening data and used to reduce false positive results [18,19,20,21].

Here, we established a public database (https://rusptools.shinyapps.io/dbRUSP) (accessed on 13 July 2022) of NBS data from a large and ethnically diverse cohort of screen-negative singleton babies reported by the California NBS program. Web-based applications were developed for interactive analysis and visualization of MS/MS-based screening data in relation to six covariates. Using dbRUSP, users can explore the breath of metabolic differences in human populations, investigate the influence of single variables and of multiple variables on metabolite levels, search for novel candidate metabolites and markers ratios for the detection of metabolic disorders, and utilize this resource for the development and validation of novel algorithms to increase the accuracy of genetic disease screening.

## 2. Materials and Methods

### 2.1. Data Summary

NBS data from 503,935 screen-negative singleton babies born between 2013 and 2017 were selected at random from the California NBS program. Babies that are screen-negative in first-tier NBS are reported as negative with no additional testing. The data included 41 metabolic analytes measured by MS/MS [22] and six covariates including the continuous variables of birth weight, gestational age, and age at blood collection and the categorical variables of sex, parenteral nutrition (TPN) status, and parent-reported ethnicity. We removed 3396 newborns with a birth weight of less than 1000 g or larger than 5000 g; gestational age of less than 28 weeks or more than 42 weeks; age at blood collection of less than 12 h or more than 168 h; or unknown information, which resulted in 500,539 newborns included in the database. Newborns were divided into each of five GA and BW groupings. The five GA categories included preterm (28–36 weeks), early-term (37–38 weeks), full-term (39–40 weeks), late-term (41 weeks), and post-term (42 weeks) birth [23]. The five BW categories included low BW (1000–2499 g), high BW (4001–5000 g), and normal BW [24], which was divided into three normal BW categories of 2500 to 3000 g, 3001 to 3500 g, and 3501 to 4000 g. The age at blood collection was divided into 4 categories: 12–23 h, 24–48 h, 49–72 h, and 73–168 h. Blood specimens in the California NBS program may be collected as early as 12 h [25], while most other states may recommended collection between 24 and 48 h [26].

The race/ethnicity status of newborns was self-reported by the parents according to the following 17 groupings: Asian East Indian, Black, Cambodian, Chinese, Filipino, Guamanian, Hawaiian, Hispanic, Japanese, Korean, Laotian, Middle Eastern, Native American, Other Southeast Asian, Samoan, Vietnamese, and White. Of the 500,539 newborns, 89,119 (17.8%) were reported with more than one ethnicity, while 10,669 (2.1%) had an unknown status. We also categorize newborns in the 17 groupings into 4 major ethnicity groupings: Asian (Asian East Indian, Cambodian, Chinese, Filipino, Guamanian, Hawaiian, Japanese, Korean, Laotian, Other Southeast Asian, Samoan, and Vietnamese), Black, Hispanic, and White (Middle Eastern and White). Newborns reported with more than one of the four ethnicity groupings were classified according to NBS program guidelines [27]: (a) Hispanic, if reported Hispanic and any other ethnicity; (b) Black, if reported Black and any other ethnicity, except Hispanic; (c) Asian, if reported Asian and any other ethnicity, except Hispanic and Black; (d) White, if reported White only.

### 2.2. Web-Based Database and Statistical Analysis

The dbRUSP web-based applications were developed with R 4.1.1 [28] using the following R packages: shiny, ggplot2 [29], effsize [30], and ComplexHeatmap [31]. This framework facilitates the interactive evaluation of metabolite data in relation to covariates. Users can perform a variety of searches and analyses in two separate modules. The first module enables the analysis of metabolite levels in relation to individual variables. For example, users can examine metabolite level differences between male and female newborns to explore the association between sex and metabolites. Five individual-variable panels are provided for the six variables analyzed in dbRUSP. GA and BW were combined into one panel based on a high correlation between the two variables (correlation coefficient = 0.5). The second module under the “Multiple Comparisons” panel was designed for the analysis of the combined effects from multiple variables on metabolite levels. Users can compare metabolite level ranges for newborns in specific BW, GA, and ethnicity categories to the “common” metabolite ranges (e.g., BW 2500–4000 g, GA 37–41 weeks, Aabc 24–48 h, no TPN) computed for newborns in the larger “reference” group (*n* = 305,674). dbRUSP provides users with detailed summary reports for each group comparison, which include the variable ranges for the selected and the common group and the mean, median, and Cohen’s d [32] for the selected metabolites.

## 3. Results

dbRUSP comprises an online database and tools for the analysis of MS/MS-based metabolic screening data and covariates. This first version (v1.0), reported in this paper, was built for a cohort of 500,539 screen-negative newborns reported by the California NBS program. The database contains two modules to study the influence on metabolite levels from individual variables and the joint effects from multiple variables.

### 3.1. Module 1: Influence of GA and BW on Metabolite Levels

GA and BW are highly correlated and can be analyzed together in one panel under the first module. As shown in Figure 1, this panel is divided into two parts with an operating part (input, left side) and a results part (output, right side). Users can select metabolic analytes or analyte ratios (Figure 1a, C3 shown as an example) and include cohorts based on selected variables. Selections can be based on ethnicity (major or detailed grouping), age at blood collection (24–48 h), sex (male and female), and with or without TPN (Figure 1c). After clicking submit, the results are shown on the right (Figure 1d–f).

Figure 1d shows the median C3 levels for the study cohort divided into 25 groups defined by BW and GA. C3 levels decrease with increasing GA for each of the BW groups, while C3 levels increase with increasing BW in each of the GA groups. An exception to this trend was found for post-term newborns (GA of 42 weeks) in two BW groups (1000–2499 and 2500–3000 g). C3 median levels in these two groups showed a different trend, which could in part be due to a low BW in relation to GA (BW is less than the 10th percentile for GA) and the small sample size (*n* = 2 and *n* = 56).

The correlation between metabolite levels and GA and BW can be further explored in the output panels shown in Figure 1e,f. Here, C3 levels generally decreased with increasing GA for all newborns and newborns in the three normal BW categories, while C3 levels increased with increasing BW for newborns in the early-term, full-term, and late-term groups. However, when including all newborns (GA between 28 and 42 weeks) in the analysis, an initial decrease of C3 levels was observed for newborns with a smaller BW of 2000–3000 g compared to increasing C3 levels in newborns with a BW between 3000 and 4500 g. We previously reported an association of preterm birth (28–36 weeks) with lower BW and increased C3 levels, suggesting that GA is a stronger covariate compared to BW [10]. These results highlighted the inverse correlation for these two covariates on metabolite levels and illustrate our approach for combining GA and BW into a single panel to stratify the analysis. While Figure 1d contained five BW and five GA groups in our analysis, Figure 1e,f only included three GA and three BW groups due to the smaller sample size and large variation for the other two GA and BW groups, respectively. Estimates for these two groups would be difficult to visualize and interpret. In our dataset, the 95% CI for the BW group of 3501–4000 g (Figure 1e) and the GA group of 41 weeks (Figure 1f) are comparatively large at both sides of the figures.

### 3.2. Module 1: Influence of Ethnicity, Sex, Age at Blood Collection, and TPN on Metabolite Levels

Parent-reported ethnicity status, sex, age at blood collection, and TPN can each be analyzed individually in relation to metabolite levels in four separate panels under the first module. The panels for these variables have similar functions with the ethnicity-related analysis shown as an example in Figure 2. This panel is divided into two parts with an operating part (input, left side) and a results part (output, right side). Users can select metabolic analytes or analyte ratios (Figure 2a, C3 shown as an example) and include cohorts in the analysis based on selected variables. The cohort selected in Figure 2b includes both male and female newborns of Asian East Indian, Cambodian, Chinese, Filipino, Japanese, Korean, Laotian, or Vietnamese ancestry, with a BW between 2500 and 4000 g, GA between 37 and 41 weeks, AaBC between 24 and 48 h, and without TPN.

In this example, BW was selected for comparing different BW categories for the eight ethnicity groupings (Figure 2c). If “No Comparison” is selected as the default, metabolite levels will be compared only between the selected ethnicity groupings without further stratification. After clicking submit (Figure 1d), results on the right show boxplots of the distribution of the C3 level in the different BW categories for each ethnicity grouping (Figure 2e) and a corresponding table of their mean and median C3 levels (Figure 2f). In this comparison, Japanese ancestry newborns with a BW between 2500 and 3000 g are identified with the lowest C3 levels (mean 1.48 μmol/L), while the highest C3 levels (mean 2.18 μmol/L) are found in newborns of Asian East Indian ancestry with a BW between 3501 and 4000 g.

### 3.3. Module 2: Multiple Comparisons Analysis

dbRUSP’s second module was designed for the analysis of joint effects from multiple variables on NBS metabolic levels. Users can select newborns based on specific covariate selections and compare differences in their metabolite levels to the “common” metabolite ranges (e.g., BW 2500–4000 g, GA 37–41 weeks, Aabc 24–48 h, no TPN) among newborns in the “reference” group. Similar to Module 1, there are two parts in this panel: operating part (left part of panel) and results part (right part of panel). Users can select metabolic analytes or analyte ratios (Figure 3a, Valine, Citrulline, and C3 as examples here), include cohorts based on variables selected in the operating part (Figure 3b), and click submit for analysis (Figure 3c). In the results part, users can inspect parallel boxplots of the distribution of metabolite levels between the selected cohort and the common groups (Figure 3d) and summary reports for each group comparison, which include the mean, median, and Cohen’s d for the selected metabolites and covariate ranges for the selected and the common group (Figure 3e). Compared to the common reference group, newborn Black males with a BW between 3501 and 4000 g, GA between 39 and 40 weeks, age at blood collection between 73 and 168 h, and no TPN have significantly higher Valine levels (Cohen’s d = 1.84, 95% CI 1.55 to 2.12), slightly lower C3 levels, and no significant differences in Citrulline levels. Elevated Valine levels are associated with maple syrup urine disease (MSUD), which could lead to false-positive screening results. C3 is a blood biomarker for methylmalonic acidemia (MMA), and physiologically lower C3 levels at or below the screening cutoff could potentially lead to false negative results.

## 4. Discussion

Improved second-tier analytical tools are needed to reduce the number of false positive newborn screens for inborn metabolic disorders on the RUSP [3,17,33]. Here, we present dbRUSP, an interactive database for the analysis and interpretation of newborn screening data and for studying the effects of covariates on blood metabolite concentrations. This first version of dbRUSP (v1.0) incorporates data for 41 metabolic analytes detected by tandem-mass-spectrometry-based dried blood spot screening and six variables recorded at the time of birth (BW, GA, sex, age at blood collection, parent-reported ethnicity, TPN status) for a cohort of 500,539 screen-negative newborns reported by the California NBS program. Through web-based interactive tools, users can systematically investigate the influence of individual variables (Module 1) and the joint effects of multiple variables (Module 2) on metabolite levels.

Module 1 consists of different panels to explore metabolite level differences for selected newborn groupings based on the user input of specific covariate ranges or combinations of variables. An application of Module 1 panels is to systematically examine metabolite and metabolites ratio level differences in diverse populations, which could be used to identify false positive and false negative screens associated with the effects of covariates. For example, elevated C3 levels identified for newborns of Asian East Indian ancestry could be associated with an increased risk of false positive screens for MMA compared to newborns in other populations (Figure 2e,f). dbRUSP could also be utilized to flag potential false negatives for case review. C3/C2 levels, a screening marker ratio for detecting MMA and PA, decreases after an AaBC of 120 h [12]. This could explain the discrepancy in testing of babies affected by MMA or PA, where initial testing showed a positive result, while a second confirmatory test several days later showed a negative result [34]. These findings highlight that covariates such as AaBC and ethnic diversity in populations should be considered when establishing C3 reference ranges and screening cutoffs.

Module 2 of dbRUSP provides an interactive “Multiple Comparisons” panel for the analysis of the combined effects from multiple variables on metabolite levels. Users can select covariates based on information available for a neonate (i.e., BW, GA, sex) and generate metabolite boxplots from covariate-matched cohorts in the database. Users can generate single or multiple metabolite boxplots and compare them to the corresponding metabolite boxplots for newborns in the “reference” group. Showing metabolite boxplots side-by-side for the selected and the reference group can be used to compare metabolite level ranges between the two groups, assess summary statistics to determine the significance of the metabolic differences, and recognize metabolite level ranges that could be outliers in the selected group. Module 2 could have utility for evaluating cases with metabolite marker levels slightly below the thresholds for a RUSP disorder. For example, if C3 levels of 6.3 μmol/L or higher have been established to identify screen-positives for MMA or PA, an individual with a C3 of 6.2 μmol/L and covariate ranges similar to those shown for the selected group in Figure 3b could be associated with a false negative screen.

dbRUSP could be of significant interest to NBS reference laboratories and health care providers who routinely evaluate MS/MS data for screenable metabolic disorders. The database has utility for evaluating first-tier screen-positives and, in particular, cases with metabolite concentrations slightly above or below the established thresholds. However, users should be cautious when interpreting data from an individual baby and especially for cases from smaller cohorts associated with larger confidence intervals, such as those with a less common ethnicity or a lower BW or earlier GA (Figure 1). Additionally, any predictions should always be considered only in conjunction with established second-tier confirmatory analysis using biochemical and DNA testing of all screen-positive cases. dbRUSP data analysis and interpretation can be obtained rapidly, within minutes, given that the analytes studied are among the 41 metabolites in the database. Ratios for the 41 metabolites are instantly computed. To assess the association of metabolite markers with a specific disorder on the RUSP, a “Metabolite to condition” table is available to look up metabolites and the corresponding condition(s) with hyperlinks to the Online Mendelian Inheritance in Man (OMIM) [35]. Additionally, dbRUSP could also be used for the discovery research of novel metabolic screening markers. For example, if a proposed candidate marker shows high variability in relation to a covariate (e.g., C14/C5 with large differences between male and female), it may not have broad utility in screening or different cutoffs based on the covariate would have to be identified. Furthermore, dbRUSP may have utility for considering trends reflected in the covariates applied to the group rather than the individual. For example, users can compare the early sampling group (i.e., AaBC at 12–23 h) to later sampling groups to identify differences that could affect screening performance for some metabolic disorders [12], which can help inform screening policy guidelines. The open-source format encourages innovative uses of the database and web-based tools and facilitates the development of novel data mining algorithms that incorporate the influences of covariates on metabolism to improve accuracy in genetic disease screening. No contribution of new data is required to access or operate dbRUSP.

The availability of data from only one state’s NBS program is a current limitation of dbRUSP. Although different NBS laboratories may use different protocols and standards to process dried blood spot specimens, the metabolite-covariate relationships identified using the California NBS program data should be applicable to other state programs. However, one major exception is the parent-reported ethnicity data. Metabolic differences between Black and White newborn groupings in California may not be directly comparable to differences between Black and White babies born in other world geographies. Even within the U.S., ancestry-related metabolic differences found in dbRUSP may not be translatable to other NBS programs. For example, ancestors of most Hispanic infants born in California came from Mexico and neighboring Central American countries or territories, while Hispanic infants born in Florida may have ancestors with Cuban, Columbian, or Puerto Rican origins [36]. Additionally, family ancestry and the level of population admixture is often poorly understood, and combining individuals from diverse populations into a few broad categories (e.g., “Hispanic”) disregards a multitude of cultural and ancestral identities [37]. Future efforts to incorporate screening data and parent-reported ethnicity information from NBS programs in different states and geographic regions could shed new light on metabolic differences in newborns in diverse populations.

NBS laboratories cannot accurately compare MS/MS-based screening results due to different testing modalities. To address this major limitation, new efforts are geared towards the harmonization and standardization of newborn screening and analytics [38,39,40]. Demographic differences within diverse populations across the U.S. and worldwide represent major challenges for harmonizing newborn screening, which will require building multi-state analytics that is unbiased with respect to covariates. The dbRUSP database and web-based tools are flexible and can be customized and expanded to support collaborative efforts across NBS programs worldwide [41].

## Figures and Tables

**Figure 1 IJNS-08-00048-f001:**
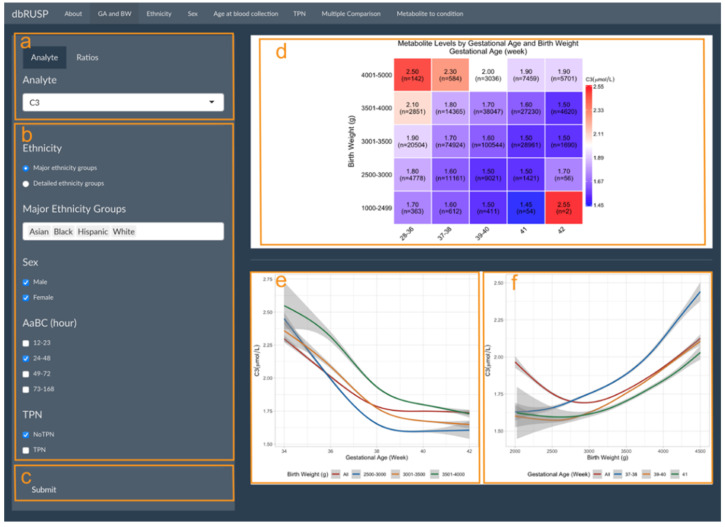
Metabolite levels by gestational age and birth weight. Module 1 panels are provided to select (**a**) metabolic analyte(s) or ratios, (**b**) covariates, and (**c**) submit criteria for analysis. (**d**) Heatmap of median C3 levels (μmol/L) in different GA (in weeks) and BW (in g) groups with the cohort size (*n*=) shown for each group. Smoothed lines display the correlation between (**e**) C3 and GA for all newborns and for newborns in the three BW groups and (**f**) C3 and BW for all newborns and for newborns in the three GA groups. Grey areas show the 95% confidence interval (CI) of the estimation.

**Figure 2 IJNS-08-00048-f002:**
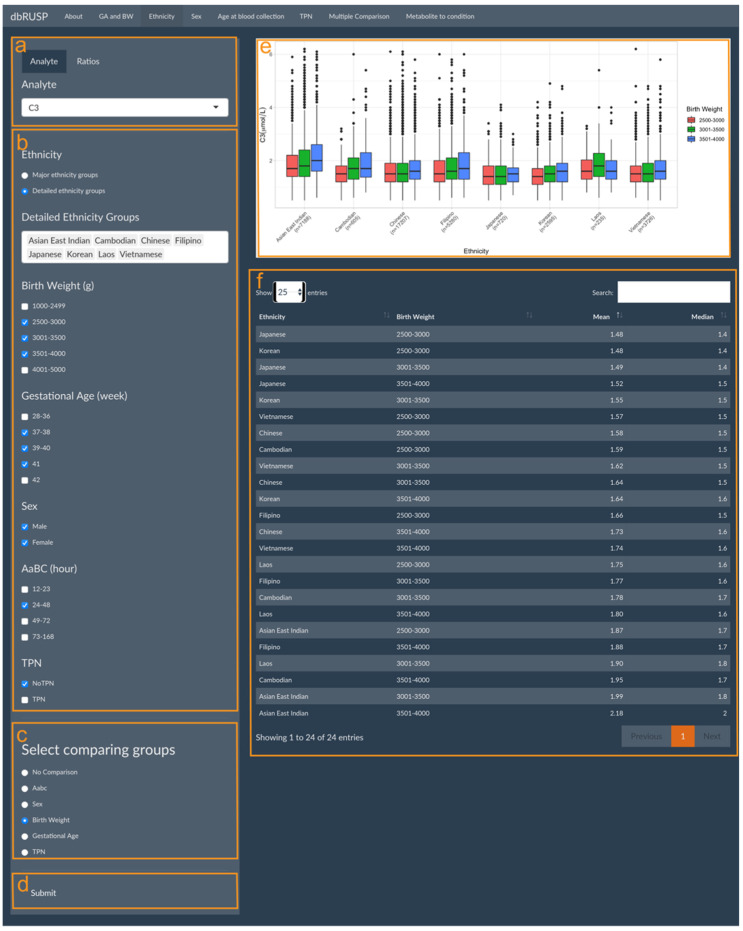
Metabolite levels in relation to parent-reported ethnicity. Module 1 panels are provided to select (**a**) metabolic analyte(s) or ratios, (**b**) covariates, (**c**) covariates for additional sub-grouping within each ethnicity group (or “No Comparison”), and (**d**) submit criteria for analysis. The results panel shows (**e**) boxplots of C3 levels in the different BW (g) categories for each ethnicity grouping and (**f**) a corresponding table of the mean and median C3 levels in the selected groups.

**Figure 3 IJNS-08-00048-f003:**
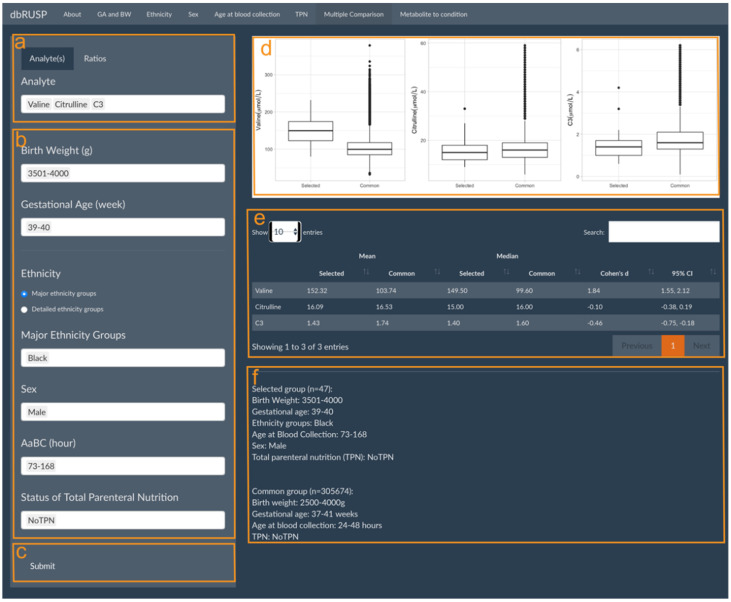
Multiple Comparisons panel under the second module. Panels are provided to select (**a**) metabolic analyte(s) or ratios, (**b**) covariates to choose from in the selected group, and (**c**) submit button to retrieve results. The results panel shows (**d**) parallel boxplots to compare Valine, Citrulline, and C3 levels between the selected and the common reference group, (**e**) a corresponding report of mean and median metabolite values, and (**f**) a description of the selected and common reference groups.

## Data Availability

The data analyzed in this study are subject to the following licenses/restrictions: The data used in this study were obtained from the California Biobank Program (CBP) under SIS Request 886. The California Department of Public Health is not responsible for the results or conclusions drawn by the authors of this publication. The data can be obtained by others after submitting a new request to the CBP coordinator. Requests to access these datasets should be directed to CaliforniaBiobank@cdph.ca.gov.

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
