# Peer review of "dbRUSP: An Interactive Database to Investigate Inborn Metabolic Differences for Improved Genetic Disease Screening"

_2409-515X, 2022, doi:10.3390/ijns8030048_

Round 1
Reviewer 1 Report
Very interesting report that will make a valuable contribution to the literature. I have no recommendations for changes.
Author Response
Thank you.
Reviewer 2 Report
The authors report the influence of six covariates on metabolite results obtained when screening just over 503,935 babies, reported as screen negative, from 2013-2017 as part of the California NBS Programme. They suggest that providing this data analysis in an open source format may help NBS Screening Labs when evaluating screen positive results that are slightly above or below established thresholds where the results could be influenced by one of the covariates reported. They suggest that this further scrutiny could help avoid both false negative and false positive screening results.
The paper is valuable and topical but the results would need to be viewed with caution when applied to an individual child and this caution, while referred to in the Discussion, could be strengthened.
In particular, one of the covariates is parent reported ethnicity, this is notoriously unreliable as family ancestry is often poorly understood within families and this should be noted. In addition, while the data collection at more than 500,000 individuals is large, when subdivided by the six groups it will leave some of the ‘rarer’ groups, such as those with less common ethnicity or at an earlier gestational age, with relatively little data and this is reflected in the wide confidence intervals shown in some of the charts, again this should perhaps be emphasised in the Discussion, in case those reading the paper were tempted to apply these concepts when interpreting results from an individual child.
The data may prove particularly useful in considering trends reflected in the covariates applied to the group rather than the individual. So for instance the authors mention (p4 line 70) that California is somewhat unusual in permitting blood collection as early as 12h after birth. As the co-variate data related to age at blood collection includes samples taken at 12-23h, 24-48h, 49-72h and 73-168h, is available from the study, it would be very interesting to report the data corresponding to those sampled at 12-23h when compared with later sampling groups. This kind of trend data may be useful in determining screening policy rather or in addition to analysis affecting the interpretation in an individual child. I would suggest that he authors report the age at sampling group data and refer to the value of the data when setting or questioning screening policy as part of the Discussion.
In general, the paper is well written, of appropriate length and employs relevant analytic tools. Once the minor additions and qualifications mentioned have been included I would recommend publication of this useful report.
Author Response
We thank the reviewer for these suggestions.
We added this sentence on p7, line 264:
However, users should be cautious when interpreting data from an individual baby and especially for cases from smaller cohorts associated with larger confidence intervals such as those with a less common ethnicity or a lower BW or earlier GA (Figure 1).
We edited this sentence on p8, line 295:
Additionally, family ancestry and the level of population admixture is often poorly understood, and combining individuals from diverse populations into a few broad categories (e.g., “Hispanic”) disregards a multitude of cultural and ancestral identities [37].
We added this sentence on p7, line 278:
Furthermore, dbRUSP may have utility for considering trends reflected in the covariates applied to the group rather than the individual. For example, users can compare the early sampling group (i.e., AaBC at 12-23 hrs) to later sampling groups to identify differences that could affect screening performance for some metabolic disorders [12], which can help inform screening policy guidelines.
Reviewer 3 Report
This is a well written manuscript which is topical and will be of interest to readers.
Typo p7, line 234. Where not were.
Typo p7 line 241 f.e., BW, GA, sex - should be i.e.,
Typo p7 line 276 laboratories
Author Response
All typos corrected. Thank you.